# Top Surface Roughness Modeling for Robotic Wire Arc Additive Manufacturing

**Heping Chen** [1,*], **Ahmed Yaseer** [1] **and Yuming Zhang** [2]

1   Ingram School of Engineering, Texas State University, San Marcos, TX 78666, USA; a_y95@txstate.edu
2   Department of Electrical and Computer Engineering, University of Kentucky, Lexington, KY 40506, USA; yuming.zhang@uky.edu
*   Correspondence: heping.chen@txstate.edu

**Abstract:** Wire Arc Additive Manufacturing (WAAM) has many applications in fabricating complex metal parts. However, controlling surface roughness is very challenging in WAAM processes. Typically, machining methods are applied to reduce the surface roughness after a part is fabricated, which is costly and ineffective. Therefore, controlling the WAAM process parameters to achieve better surface roughness is important. This paper proposes a machine learning method based on Gaussian Process Regression to construct a model between the WAAM process parameters and top surface roughness. In order to measure the top surface roughness of a manufactured part, a 3D laser measurement system is developed. The experimental datasets are collected and then divided into training and testing datasets. A top surface roughness model is then constructed using the training datasets and verified using the testing datasets. Experimental results demonstrate that the proposed method achieves less than 50 μm accuracy in surface roughness prediction.

**Keywords:** industrial robot; wire arc additive manufacturing (WAAM); roughness



## 1. Introduction

Wire Arc Additive Manufacturing (WAAM) is an additive manufacturing process for fabricating metal parts which has been used in many production processes in industries such as automotive, aerospace, nuclear, oil and gas, shipbuilding, and heavy equipment manufacturing [1–3]. Controlling the surface roughness of the manufactured parts is one of the challenging problems in WAAM processes. The surface roughness evaluation can be categorized into the top surface roughness evaluation [4,5] and side surface roughness evaluation [6,7]. Both are important for WAAM because the roughness will affect the properties of the final product. Yehorov et al. [7] evaluated the side surface waviness of thin walls produced by varying the current, wire feed speed, and travel speed. The experimental results show that a suitable range of travel speed can mitigate the roughness of the produced surfaces. Li et al. [8] developed a WAAM process to manufacture a thin-walled structure. The side surface roughness is about 5 μm, which is much better than that of other research methods. However, the proposed technique used small-power metal fine wire feeding, which is not feasible for most additive manufacturing applications when manufacturing large parts.

The top surface roughness or waviness is also worth investigating [1,4]. As we know, the top surface roughness is affected by the deposition pattern and welding process parameters. In order to improve the fabrication efficiency and accuracy, industrial robots have been widely used in the WAAM processes [9]. Various robot path planning methods such as weaving, zigzag, raster, and Medial Axis Transformation have been used in WAAM [10]. Using these different path patterns, the roughness of the produced layers is different. Typically, machining is needed to reduce the surface roughness [1,4]. Most researchers model the weld bead using a parabolic function for a straight path [4,11] and reasonable results are achieved. Aldalur etc. [12] compared the performance of an overlapping path with that

of a weaving path and they found that the weaving path achieved better surface roughness; they also mentioned that similar mechanical properties including tensile strength, hardness and Charpy were obtained. Therefore, we will investigate the surface roughness for WAAM using a robotic weaving path. Besides the robot welding path, WAAM process parameters will also affect the surface roughness [4]. In order to reduce the machining cost and improve WAAM efficiency and properties of the final product, the top surface roughness must be minimized during the WAAM processes. Thus, robot path parameters and WAAM process parameters should be investigated in order to minimize the surface roughness.

In order to control the surface roughness, a model which establishes a relationship between the robotic WAAM parameters (path and process parameters), and top surface roughness must be constructed.

The Response Surface Methods have been implemented in the top surface roughness modeling [13,14]. However, it is difficult for these methods to accurately model the complex welding process to predict the surface roughness because only polynomial fittings are used. Recently machine learning methods have been implemented to model top surface roughness. Different machine learning methods have been proposed [4,15,16]. Xia et al. [16] explored surface roughness modeling and prediction methods based on Adaptive Neuro-fuzzy Inference System (ANFIS), Extreme Learning Machine and Support Vector Regression. They compared the performances of these methods and found that ANFIS achieved better results in terms of mean absolute percentage error (MAPE) which is about 14.15%. Yaseer et al. [5] proposed a machine learning method based on Random Forest to model the surface roughness. Their results show that the MAPE is 5.64%. Even though this method achieves better MAPE, the computation is quite complex. Therefore, more investigation about predicting the top surface roughness with high accuracy is needed, especially for robotic WAAM processes using a weaving path.

Machine learning methods based on Gaussian Process Regression (GPR) have been widely used to model some very complex processes in different applications [17–20]. These methods can deal with system uncertainties and noisy observations in complex system modeling [21]. Therefore, it is desirable to develop a modeling method based on GPR for complex WAAM roughness modeling.

In this paper, we propose a machine learning method based on GPR to establish a relationship between the robotic WAAM parameters and produced surface roughness using a weaving path. The procedures to collect training and testing datasets are detailed. A surface roughness measurement method is presented. After the model based on GPR is constructed, the testing datasets are used to test the accuracy of the generated model. The experimental results are presented, and analysis is detailed. The results demonstrate that the proposed machine learning method based on GPR achieves about 50 μm accuracy in WAAM surface roughness modeling. The main contributions of this paper are:

- A machine learning method based on GPR is developed to model the top surface roughness for robotic weaving path;
- The proposed method is tested experimentally, and the results demonstrate that about 50 μm accuracy of the top surface roughness is achieved.

## 2. Materials and Methods

WAAM is a very complex process and it is difficult to predict the surface roughness. Therefore, a method based on GPR is proposed to establish the relationship between the robotic WAAM parameters and surface roughness. An industrial robot is programmed to perform the WAAM process using a weaving path. A 3D measurement system is used to collect the surface dimensional data. Using the collected datasets, a model based on GPR is then constructed.

### 2.1. Robotic WAAM Process

For robotic WAAM processes, there are two major problems: robot trajectory planning and WAAM process parameter control. Different methods have been developed for

robotic WAAM path generation such as raster, grid, zigzag, contour offset, spiral, and weaving [22–24]. In this paper, we will study the roughness for robotic WAAM process using a weaving path as shown in Figure 1.

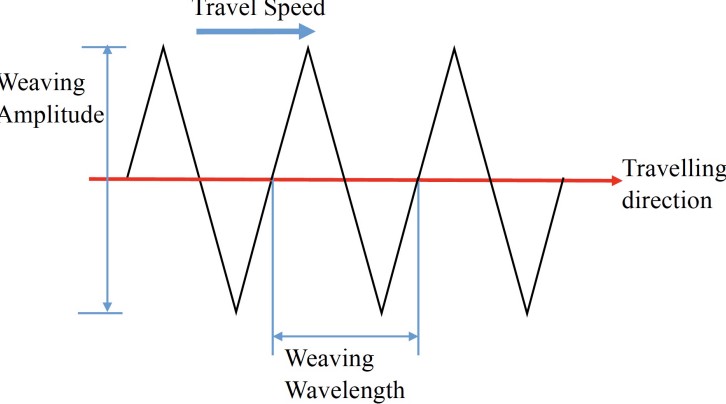

**Figure 1.** A robotic weaving path. There are three trajectory parameters: travel speed, weaving wavelength, and weaving amplitude.

As we can see in Figure 1, there are three parameters to be determined: travel speed, weaving wavelength, and weaving amplitude. These parameters will affect the surface roughness. The WAAM process parameters include wire feed speed, welding current, welding voltage, wire electrode size, type of shielding gas, electrode extension and electrode angle, etc. [25]. In this paper, only four parameters are considered: wire feed speed $F_S$, travel speed ($T_S$), weaving wavelength ($W_W$), and weaving amplitude ($W_A$). Other parameters are fixed in the experiments.

*2.2. Data Collection and Roughness Measurement*

For a robotic WAAM process, the input parameters can be defined as:

$$\boldsymbol{\theta} = [F_S \ T_S \ W_W \ W_A] \tag{1}$$

where $\boldsymbol{\theta}$ is a vector of the robotic WAAM parameters. The surface roughness $R$ can be modeled using the input parameters as:

$$R = \rho(\boldsymbol{\theta}) + \nu \tag{2}$$

where $\nu \sim \mathcal{N}(0, \sigma)$ is Gaussian noise with mean 0 and variance $\sigma$. $\rho(\boldsymbol{\theta})$ is a Gaussian Process. Constructing a physical model for a robotic WAAM process to predict the roughness is almost impossible because the complexity of the system which a includes robotic system and a welding system, etc. However, we can collect experimental data to construct a model using machine learning methods. By performing experiments, we can obtain the experimental data $(\boldsymbol{\theta}, R)$ in Equations (1) and (2). The input vector $\boldsymbol{\theta}$ can be set before experiments and recorded. After an experiment is performed, a part can be fabricated. A fabricated part surface is shown in Figure 2.

In order to measure the surface roughness shown in Figure 2, a 3D laser scanning system is used. After the surface is scanned, point cloud data can be collected. The collected point cloud data must be processed in order to obtain the top surface roughness. Because the 3D laser scanning system measures a larger area than the top surface area of a fabricated part, the extra point cloud data must be removed. In order to explain the method we developed to obtain the top surface roughness, we use an example of the collected point cloud data of a fabricated part surface as shown in Figure 3.

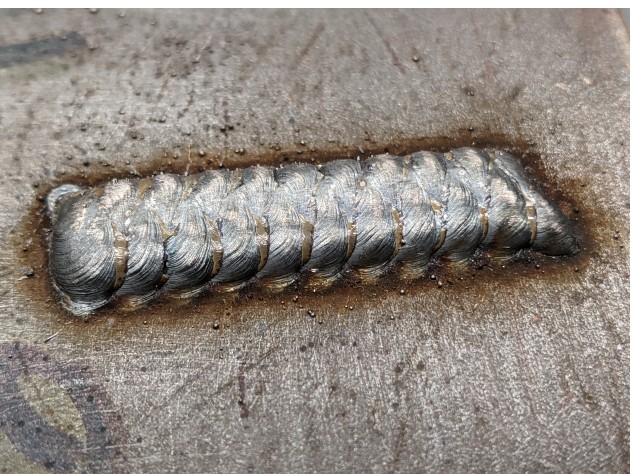

**Figure 2.** A fabricated part on a substrate.

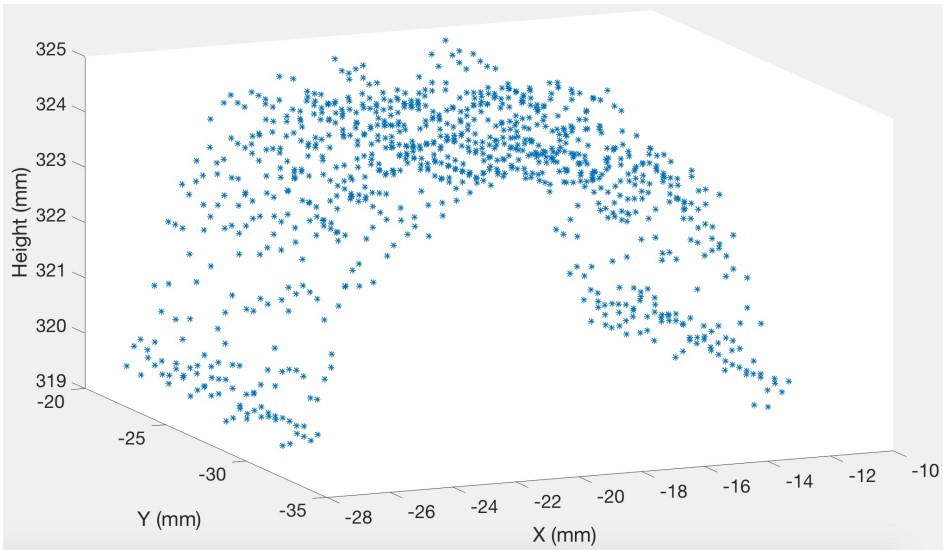

**Figure 3.** One example of the measured point cloud data of a fabricated part. The height is measured relative to the frame of the 3D laser scanning system. * in the figure represents data points.

In order to remove the extra data points, we fit several planes to the point cloud data. Because the top surface roughness is computed using data points on the top surface, the edge data in the point cloud should be removed first. We fit a plane to the point cloud data and remove all data points below the plane. Using the data points above the plane, another plane is fitted again and the data points below the plane are removed again. The process continues until all edge data points are removed. Using all data points above the final plane, a final plane is fitted to find the top surface roughness. Suppose the fitted plane is:

$$ax + by + cz + 1 = 0 \tag{3}$$

where $a$, $b$, $c$ are coefficients and $x$, $y$, $z$ are coordinates. The roughness is then calculated using the following equation:

$$R = \frac{1}{K} \sum_{k=1}^{K} \frac{|ax_k + by_k + cz_k + 1|}{\sqrt{a^2 + b^2 + c^2}} \tag{4}$$

where $K$ is the total number of points used to calculate the roughness; $(x_k, y_k, z_k)$ are the coordinates of the $k^{th}$ point.

After $n$ experiments are performed, $n$ sets of experimental data $(\boldsymbol{\theta}, \mathbf{R})$ can be obtained. These $n$ datasets can be used as training datasets:

$$
\begin{aligned}
\boldsymbol{\theta} &= [\boldsymbol{\theta}_1, \boldsymbol{\theta}_2, ..., \boldsymbol{\theta}_n] \\
\mathbf{R} &= [R_1, R_2, ..., R_n]
\end{aligned}
\tag{5}
$$

### 2.3. Modeling Method

Once the experimental datasets are obtained, we can construct a model between the input parameters and output surface roughness. GPR is a machine learning method which has been used to construct a model for a complex process in many applications. In Equation (2), $\rho(\boldsymbol{\theta})$ is defined using mean and covariance function:

$$
\rho(\boldsymbol{\theta}) \sim \mathcal{GP}(m(\boldsymbol{\theta}), k(\boldsymbol{\theta}, \boldsymbol{\theta}'))
\tag{6}
$$

where $\mathcal{GP}$ represents the Gaussian Process; $\boldsymbol{\theta}'$ is a set of arbitrary input WAAM parameters; $m(\boldsymbol{\theta})$ and $k(\boldsymbol{\theta}, \boldsymbol{\theta}')$ are mean and covariance function, respectively.

The covariance function $k(\boldsymbol{\theta}, \boldsymbol{\theta}')$ is defined as:

$$
k(\boldsymbol{\theta}, \boldsymbol{\theta}') = \mathbf{E}[(m(\boldsymbol{\theta}) - R(\boldsymbol{\theta}))(m(\boldsymbol{\theta}') - R(\boldsymbol{\theta}'))]
\tag{7}
$$

Different covariance functions can be used for model construction as shown in Table 1.

**Table 1.** Covariance functions $k(\boldsymbol{\theta}, \boldsymbol{\theta}')$.

| Name | Equation |
|------|----------|
| Noise Function | $\sigma^2$ |
| Linear Function | $\boldsymbol{\theta} \cdot \boldsymbol{\theta}'$ |
| Squared Exponential Function | $\sigma^2 \exp\left(-\frac{|\boldsymbol{\theta} - \boldsymbol{\theta}'|^2}{2l^2}\right)$ |
| Gaussian Radial Basis Function | $\exp(-\beta|\boldsymbol{\theta} - \boldsymbol{\theta}'|^2)$ |
| Sigmoid Function | $\tanh[\beta(\boldsymbol{\theta} \cdot \boldsymbol{\theta}') + c]$ |
| Polynomial Function | $[\beta(\boldsymbol{\theta} \cdot \boldsymbol{\theta}') + c]^\alpha$ |

Note: $\sigma$, $l$, $\beta$, $c$ and $\alpha$ are hyperparameters.

The covariance functions have hyperparameters which should be determined in order to construct a model. For example, the Gaussian Radial Basis function has one hyperparameter $\beta$. Because different covariance function may have different number of hyperparameters, we denote the hyperparameters using vector $\mathbf{h}$.

The covariance functions can also be combined to generate a new covariance function for model construction; for example,

$$
k(\boldsymbol{\theta}, \boldsymbol{\theta}') = \sum_{p=1}^{P} w_p k_p(\boldsymbol{\theta}, \boldsymbol{\theta}')
\tag{8}
$$

where $k_p(\boldsymbol{\theta}, \boldsymbol{\theta}')$ is the $p^{th}$ covariance function and $w_p$ is its weight. For the combined covariance function, more hyperparameters should be determined.

For a training dataset $(\boldsymbol{\theta}, \mathbf{R})$, the posterior probability of $\rho(\boldsymbol{\theta})$ can be calculated:

$$
p(\rho|\boldsymbol{\theta}, \mathbf{R}, \mathbf{h}) = \frac{p(\mathbf{R}|\boldsymbol{\theta}, \rho, \mathbf{h})p(\rho|\boldsymbol{\theta}, \mathbf{h})}{p(\mathbf{R}|\boldsymbol{\theta}, \mathbf{h})}
\tag{9}
$$

We can then compute the marginal likelihood:

$$
p(\mathbf{R}|\boldsymbol{\theta}, \mathbf{h}) = \int p(\mathbf{R}|\boldsymbol{\theta}, \rho, \mathbf{h})p(\rho|\boldsymbol{\theta}, \mathbf{h})d\rho
\tag{10}
$$

As we can see that $p(\mathbf{R}|\boldsymbol{\theta}, \mathbf{h})$ is the likelihood of hyperparameters $\mathbf{h}$. The log marginal likelihood can then be derived:

$$\log p(\mathbf{R}|\boldsymbol{\theta}, \mathbf{h}) = -\frac{1}{2}\mathbf{R}^{\mathrm{T}}\Gamma^{-1}\mathbf{R} - \frac{1}{2}\log|\Gamma| - \frac{n}{2}\log 2\pi \tag{11}$$

where $\Gamma$ is defined as:

$$\Gamma = K(\boldsymbol{\theta}, \boldsymbol{\theta}) + \sigma^2 I \tag{12}$$

The hyperparameters can then be optimized by maximizing the log marginal likelihood:

$$\mathbf{h}^* = \underset{\mathbf{h}}{\operatorname{argmax}} \log p(\mathbf{R}|\boldsymbol{\theta}, \mathbf{h}) \tag{13}$$

After the optimal hyperparameters are obtained, the covariance function can be determined, and a roughness prediction model can be established. For a given set of input parameters $\boldsymbol{\theta}^*$, the output roughness $\mathbf{R}^*$ can be predicted. For $m$ sets of given input parameters

$$\boldsymbol{\theta}^* = [\boldsymbol{\theta}_1^*, \boldsymbol{\theta}_2^*, ..., \boldsymbol{\theta}_m^*], \tag{14}$$

let the predicted output parameters be

$$\mathbf{R}^* = [R_1^*, R_2^*, ..., R_m^*]. \tag{15}$$

The joint marginal likelihood of output parameters can be expressed as:

$$p(\mathbf{R}, \mathbf{R}^*) \sim \mathcal{N}(0, \mathbf{K}) \tag{16}$$

where

$$\mathbf{K} = \left[ \begin{array}{cc} K(\boldsymbol{\theta}, \boldsymbol{\theta}) + \sigma^2 I & K(\boldsymbol{\theta}, \boldsymbol{\theta}^*) \\ K(\boldsymbol{\theta}^*, \boldsymbol{\theta}) & K(\boldsymbol{\theta}^*, \boldsymbol{\theta}^*) \end{array} \right] \tag{17}$$

where $K(\boldsymbol{\theta}', \boldsymbol{\theta}'')$ ($\boldsymbol{\theta}'$ and $\boldsymbol{\theta}''$ represent $\boldsymbol{\theta}$ and $\boldsymbol{\theta}^*$) is a matrix defined using covariance function $k(\boldsymbol{\theta}_i', \boldsymbol{\theta}_j'')$:

$$K_{ij}(\boldsymbol{\theta}', \boldsymbol{\theta}'') = k(\boldsymbol{\theta}_i', \boldsymbol{\theta}_j'') \tag{18}$$

where $K_{ij}(\boldsymbol{\theta}', \boldsymbol{\theta}'')$ is the element in the matrix $K(\boldsymbol{\theta}', \boldsymbol{\theta}'')$.

We can then calculate the output surface roughness $\mathbf{R}^*$ using the conditional distribution:

$$p(\mathbf{R}^*|\boldsymbol{\theta}, \mathbf{R}, \boldsymbol{\theta}^*) \sim \mathcal{N}(\mu(\mathbf{R}^*), \mathrm{V}(\mathbf{R}^*)) \tag{19}$$

$$\begin{aligned} \mu(\mathbf{R}^*) &= K(\boldsymbol{\theta}^*, \boldsymbol{\theta})\Gamma^{-1}\mathbf{R} \\ \mathrm{V}(\mathbf{R}^*) &= K(\boldsymbol{\theta}^*, \boldsymbol{\theta}^*) - K(\boldsymbol{\theta}^*, \boldsymbol{\theta})\Gamma^{-1}K(\boldsymbol{\theta}, \boldsymbol{\theta}^*) \end{aligned} \tag{20}$$

where $\mu(\mathbf{R}^*)$ and $\mathrm{V}(\mathbf{R}^*)$ are the predicted mean and variance of $\mathbf{R}^*$, respectively. Therefore, given a set of input parameters $\boldsymbol{\theta}^*$, the mean and variance of the output surface roughness $\mathbf{R}^*$ can be predicted.

## 3. Results and Discussions

To validate the proposed modeling method using GPR, a WAAM system and a measurement system were developed.

Figure 4 shows the developed WAAM system which includes an ABB IRB140 robot with IRC5 controller, a welding machine (Thermal Arc PowerMaster 500 by Thermadyne Industries, Inc., West Lebanon, NH, USA), a Miller r-115 wire feeder and a shielding gas tank.

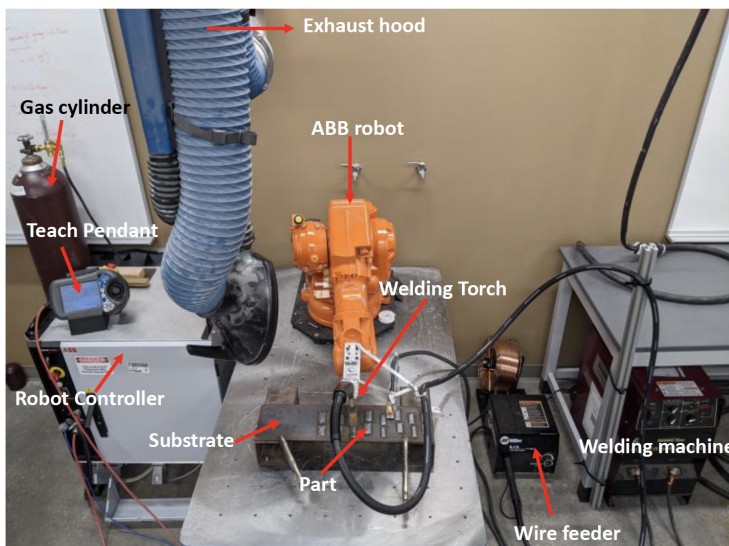

**Figure 4.** The developed experimental system.

The flow rate of the shielding gas (90% Argon and 10% $CO_2$) was 31.78 $ft^3$/h, and 1.2 mm diameter steel wire (ER70S-6) was used. Direct current reverse polarity (DCRP) was used for connecting the welding machine with the MIG gun and substrate.

Thirty trial experiments were performed using different robotic WAAM process parameters, and we found that the bead cross-section tends to be parabolic instead of flat with the increase in travel speed and wire feed speed using a weaving path. Higher wire feed speed causes more spatter. Considering these issues, we chose the values of input parameters as shown in Table 2 to perform the experiments.

**Table 2.** The range of the input parameters used in experiments.

| Input Parameters | Minimum Values | Maximum Values |
|---|---|---|
| $F_S$ (inches/minute) | 60 | 90 |
| $T_S$ (mm/s) | 3 | 5 |
| $W_W$ (mm) | 2 | 4 |
| $W_A$ (mm) | 6 | 12 |

As we can see from Table 2, there were four input parameters. The ABB robot was programmed to perform the experiments. We totally fabricated 45 parts. Figure 5 shows some examples of the fabricated parts.

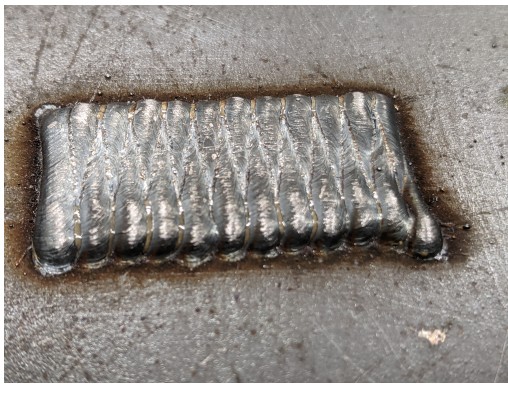

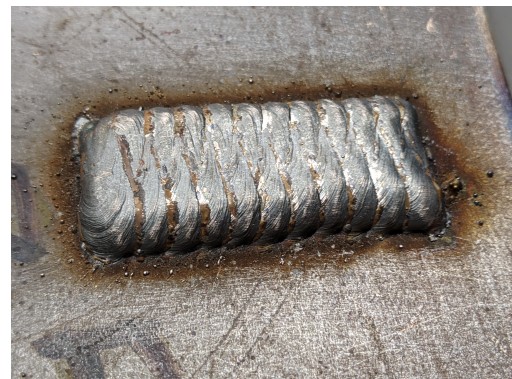

(a) $F_S = 60$; $T_S = 4.5$; $W_W = 3.5$; $W_A = 10$; $R = 0.2719$        (b) $F_S = 90$; $T_S = 5$; $W_W = 3.5$; $W_A = 9$; $R = 0.2353$

**Figure 5.** *Cont.*

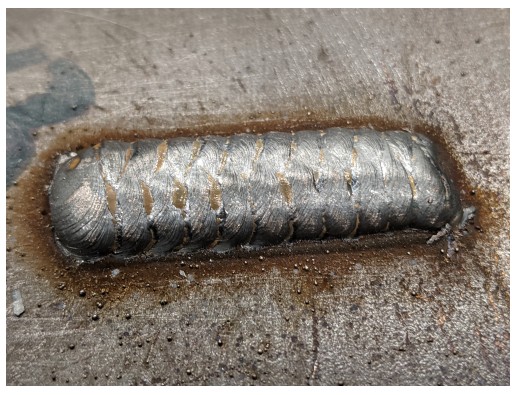 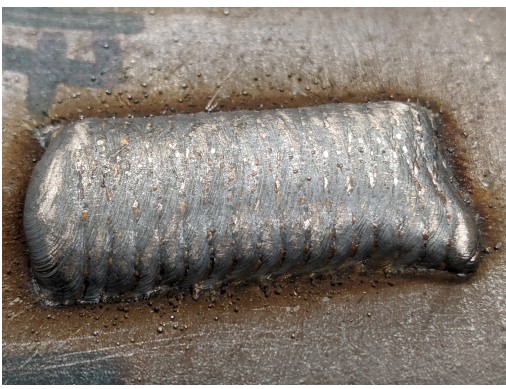

(**c**) $F_S = 75$; $T_S = 3.5$; $W_W = 3.5$; $W_A = 6$; $R = 0.3806$      (**d**) $F_S = 75$; $T_S = 5$; $W_W = 2.5$; $W_A = 10$; $R = 0.3711$

**Figure 5.** The fabricated parts using different parameters.

In order to measure the surface roughness of the fabricated parts, a measurement system is developed and shown in Figure 6.

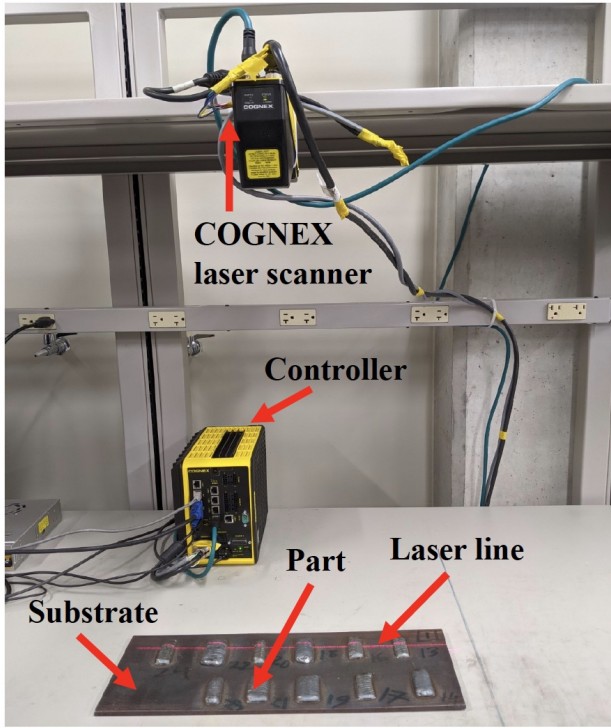

**Figure 6.** The developed measurement system for surface roughness.

The measurement system includes a Cognex 3D laser scanner with measurement accuracy 10 μm. After scanning the manufactured part, point cloud data can be obtained. We processed the data using MATLAB by fitting a series of planes. In this research, we found that fitting two planes was good enough to compute the surface roughness. Figure 7 shows the results after fitting the first plane and the second plane. The data above the first fitted plane were used to calculate the surface roughness.

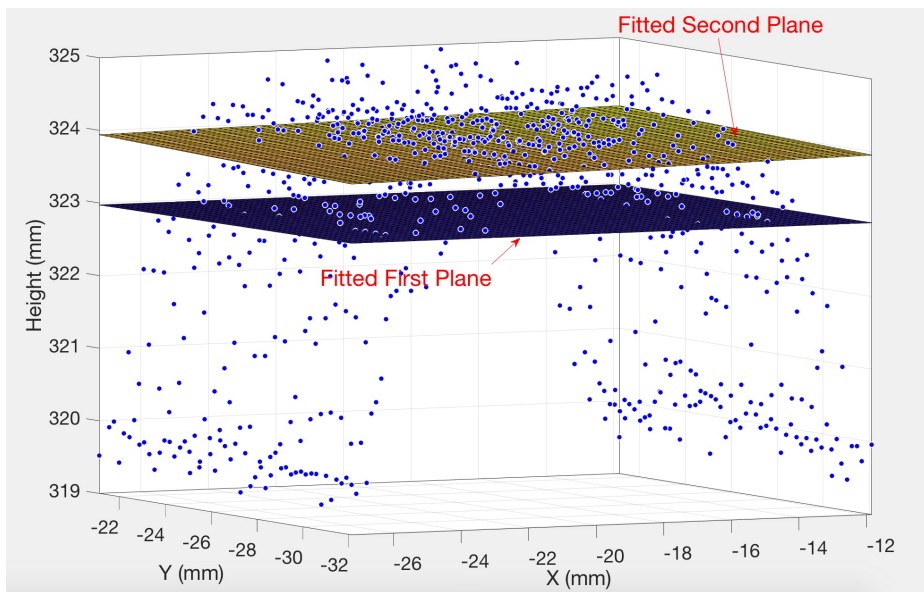

**Figure 7.** Fitted planes to the point cloud data to obtain surface roughness. The points below the plane will be removed and the points above the plane are used to fit the next plane. ● represents data points.

For the 45 parts, we collected the point cloud data and computed the surface roughness. We obtained a total of 45 datasets. Figure 8 shows one example of the relationship between the robotic WAAM parameters ($T_s$ and $F_s$) and the surface roughness. From the figure, we can see that their relationship is quite nonlinear.

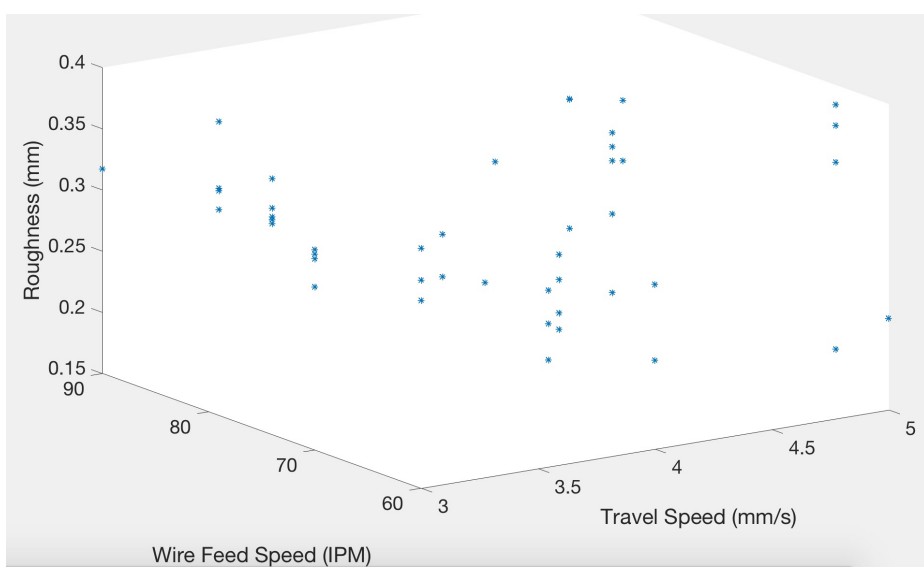

**Figure 8.** The relationship between the robotic WAAM parameters and surface roughness. * represents data points.

The learning method based on GPR was then implemented to construct a model to establish the relationship between the robotic WAAM process parameters and the top surface roughness. In the training process, we combined three covariance functions to generate a covariance function for model construction: noise, Squared Exponential Function and Gaussian Radial Basis Function. Therefore, there were four hyperparameters to be optimized. The weight $w_p$ for the three covariance functions was set to be 1.

We divided the recorded 45 datasets into 40 training datasets and 5 testing datasets randomly. We constructed a model using the training datasets based on the proposed GPR

method. We then used the five testing datasets to test the model. The software flowchart is shown in Algorithm 1.

---

**Algorithm 1** Model Construction and Testing Procedures Based on GPR.

---

**procedure**
    Read experimental datasets from a file
    Randomly divide the datasets into 40 training datasets and 5 testing datasets
    Construct a model based on GPR using the training datasets
    *loop the 40 training datasets*:
    Input the robotic WAAM process parameters in the training dataset into the constructed model and obtain the predicted top surface roughness
    Compute RMS error using the predicted top surface roughness and the actual top surface roughness
    Record the maximum top surface roughness
    **goto** *loop*.
    *loop the 5 testing datasets*:
    Input the robotic WAAM process parameters in the testing dataset into the constructed model and obtain the predicted top surface roughness
    Compute the RMS error using the predicted top surface roughness and the actual top surface roughness
    Record the maximum top surface roughness
    **goto** *loop*.
    **close**

---

Figure 9a shows one of the results of training errors of the 40 training datasets and Figure 9b one of the results of the five testing datasets.

This model construction and testing process was performed about 100 times using the randomly generated training and testing datasets and the results were recorded. Table 3 summarizes the maximum errors and root mean square (RMS) errors of the training datasets and testing datasets in four trials.

**Table 3.** The Testing Results.

| Trial | RMS1 (μm) | Max1 (μm) | RMS2 (μm) | Max2 (μm) |
|:-----:|:---------:|:---------:|:---------:|:---------:|
| 1 | 0.064 | 0.178 | 22.830 | 41.001 |
| 2 | 0.096 | 0.269 | 27.102 | 45.626 |
| 3 | 0.834 | 2.153 | 22.238 | 34.499 |
| 4 | 0.065 | 0.149 | 28.407 | 46.845 |

Note: Max1 and RMS1 are the maximum error and RMS error of the training datasets, respectively; Max2 and RMS2 are the maximum error and RMS error of the testing datasets, respectively.

As we can see in Table 3, the maximum error of the training datasets is less than 2.5 μm and the RMS error is less than 1 μm. The results mean that the modeling accuracy is very high using the training datasets. In order to demonstrate the modeling accuracy, we also used the random testing datasets to validate the model. Table 3 shows the maximum prediction error is less than 50 μm and the RMS error is less than 30 μm. The testing results demonstrate that the modeling method using GPR can accurately predict the surface roughness given a set of robotic WAAM parameters.

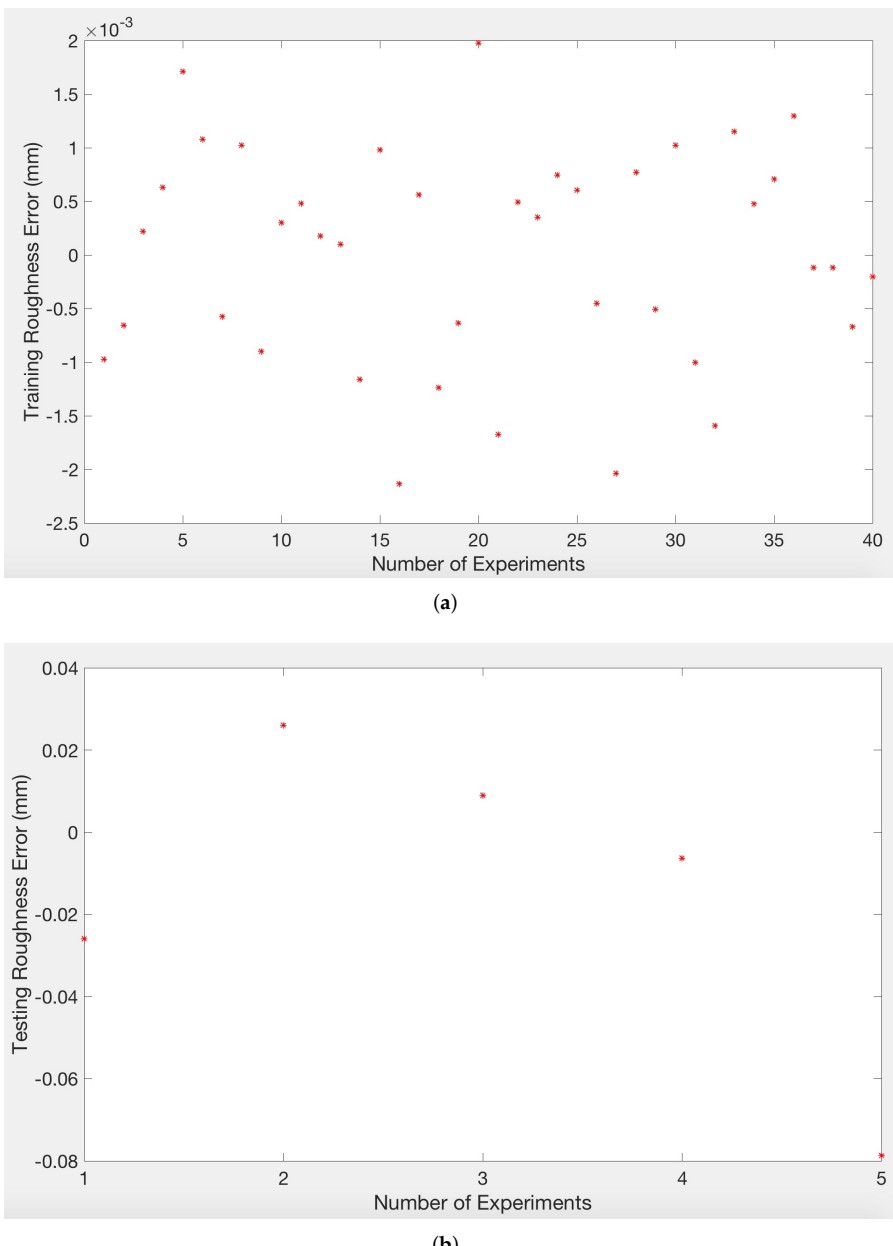

**Figure 9.** The errors of the training datasets and testing datasets: (**a**) errors of the 50 training datasets; (**b**) errors of the 5 testing datasets. * represents data points.

## 4. Conclusions and Future Work

One of the challenges in the WAAM process is to control the surface roughness, which is affected by the WAAM process parameters. A machine learning method based on Gaussian Process Regression was developed to model the relationship between the WAAM process parameters and top surface roughness. Experiments were performed and the surface roughness was measured using a 3D laser scanning system. After the datasets were collected, they were divided into the training and testing datasets. A model was then generated using the training datasets and tested using the testing datasets. The experimental results show that the maximum prediction error was less than 50 μm, which demonstrates the effectiveness of the modeling method using Gaussian Process Regression. Our future work will focus on modeling the surface height and surface width together with the top surface roughness. Once the model is created, we can explore an optimal set of WAAM process parameters such that desired surface height, surface width, and minimal

top surface roughness can be achieved, which is useful for variable layer thickness and printing width in additive manufacturing.

**Author Contributions:** Conceptualization, H.C.; methodology, H.C.; software, H.C. and A.Y.; validation, A.Y. and H.C.; formal analysis, H.C. and A.Y.; investigation, H.C.; resources, H.C. and Y.Z.; data curation, A.Y. and H.C.; writing—original draft preparation, H.C.; writing—review and editing, H.C.; visualization, H.C. and A.Y.; supervision, H.C. and Y.Z.; project administration, H.C.; funding acquisition, H.C. and Y.Z. All authors have read and agreed to the published version of the manuscript.

**Funding:** The research is partially supported by the Research Enhancement Program, Texas State University.

**Data Availability Statement:** Not applicable.

**Conflicts of Interest:** The authors declare no conflict of interest.

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
