# Peer review of "Top Surface Roughness Modeling for Robotic Wire Arc Additive Manufacturing"

_jmmp, doi:10.3390/jmmp6020039_

Round 1
Reviewer 1 Report
The manuscript "Surface Roughness Modeling for Robotic Wire Arc Additive Manufacturing" was presented for review. The paper is devoted to the essential issue of control of the surface roughness in the selected additive manufacturing process.
The authors used the algorithm of machine learning to decrease the surface roughness and to control the maximum prediction error of less than 50 microns.
There are several issues to be fixed in the paper before it can be recommended for publication.
1. Please, re-organize the manuscript in a standard MDPI structure for research papers: Introduction, Methods and Materials, Results and Discussion, and Conclusion.
2. Please, In the Introduction, formulate clearly the gap that you wanted to research, to show the proposed novelty.
3. Please, add the recent papers focused on the same topic into the state of the art section of Introduction. E.g. https://doi.org/10.1016/j.jii.2021.100218
https://doi.org/10.1051/mfreview/2020005
https://doi.org/10.1016/j.rineng.2021.100330
4. Please, re-write the Materials and Methods section, don't mix it with results. In this section, you should put only information about equipment, material, modeling parameters, without results and their interpretation.
5. Please put Table 3 to the 3rd section where it was mentioned
6. line 151 - no need to decipher the abbreviation twice "Gaussian Process Regression (GPR)"
7. line 113 "Many trial experiments were performed..." - please specify the number of experimental trials.
Author Response
Dear Editors and Reviewers:
Thank you very much for your time and effort to process and review our paper. Please check the attached file for our reply.
Sincerely yours,
Heping Chen

Reviewer 2 Report
The manuscript is well structured and written.
It is suggested to go through the manuscript to amend typos, such as space of a number and its unit.
It is suggested to change the font color of the annotations in Figs. 4 and 6.
Author Response

(The authors gave the same response as above.)

Reviewer 3 Report
The article discusses a machine learning method based on Gaussian process regression (GPR) to build a model between WAAM process parameters and the surface roughness of a manufactured part. This research topic is one of the most relevant today, and the selected research objects have a high potential for practical application. However, after reading the reviewer, a number of questions arose that the authors should answer before the work can be published.
1 In the introduction, authors should draw attention to similar studies and provide a brief overview of the current state of research in this area.
2 The authors should explain how exactly they take into account the change in roughness during the manufacturing process.
3 The used machine learning model requires more detailed explanations in terms of the algorithms used.
4 Conclusion requires significant improvement in terms of reflecting the results obtained and plans for future research.
5 The technical remarks include the poor quality of the drawings. The authors should correct the quality of the figures.
Author Response

(The authors gave the same response as above.)

Round 2
Reviewer 1 Report
The authors have modified the manuscript according to all my recommendations.
Reviewer 3 Report
The authors answered all the questions of the reviewer and significantly improved the article. The article may be accepted for publication.